# Predicting Tree Sap Flux and Stomatal Conductance from Drone-Recorded Surface Temperatures in a Mixed Agroforestry System—A Machine Learning Approach

**Florian Ellsäßer** [1,*]**, Alexander Röll** [1] **, Joyson Ahongshangbam** [1]**, Pierre-André Waite** [2]**, Hendrayanto** [3]**, Bernhard Schuldt** [4] **and Dirk Hölscher** [1,5]

[1]   Tropical Silviculture and Forest Ecology, University of Goettingen, Büsgenweg 1, 37077 Göttingen, Germany; aroell@gwdg.de (A.R.); jahongs@gwdg.de (J.A.); dhoelsc@gwdg.de (D.H.)

[2]   Plant Ecology and Ecosystems Research, University of Goettingen, Untere Karspüle 2, 37073 Göttingen, Germany; pwaite@gwdg.de

[3]   Forest Management, Kampus IPB Darmaga, Bogor Agricultural University, Bogor 16680, Indonesia; hendrayanto@apps.ipb.ac.id

[4]   Julius-von-Sachs-Institute for Biological Sciences, Chair of Ecophysiology and Vegetation Ecology, University of Wuerzburg, Julius-von-Sachs-Platz 3, 97082 Wuerzburg, Germany; bernhard.schuldt@uni-wuerzburg.de

[5]   Centre of Biodiversity and Sustainable Land Use, University of Goettingen, Platz der Göttinger Sieben 5, 37073 Göttingen, Germany

*   Correspondence: fellsae@gwdg.de

**Abstract:** Plant transpiration is a key element in the hydrological cycle. Widely used methods for its assessment comprise sap flux techniques for whole-plant transpiration and porometry for leaf stomatal conductance. Recently emerging approaches based on surface temperatures and a wide range of machine learning techniques offer new possibilities to quantify transpiration. The focus of this study was to predict sap flux and leaf stomatal conductance based on drone-recorded and meteorological data and compare these predictions with in-situ measured transpiration. To build the prediction models, we applied classical statistical approaches and machine learning algorithms. The field work was conducted in an oil palm agroforest in lowland Sumatra. Random forest predictions yielded the highest congruence with measured sap flux ($r^2 = 0.87$ for trees and $r^2 = 0.58$ for palms) and confidence intervals for intercept and slope of a Passing-Bablok regression suggest interchangeability of the methods. Differences in model performance are indicated when predicting different tree species. Predictions for stomatal conductance were less congruent for all prediction methods, likely due to spatial and temporal offsets of the measurements. Overall, the applied drone and modelling scheme predicts whole-plant transpiration with high accuracy. We conclude that there is large potential in machine learning approaches for ecological applications such as predicting transpiration.

**Keywords:** transpiration; method comparison; UAV; oil palm; multiple linear regression; support vector machine; random forest; artificial neural network

## 1. Introduction

Transpiration is the largest water flux from terrestrial surfaces, accounting for 80%–90% of terrestrial evapotranspiration [1]. Transpiration is strongly affected by changes in land cover and land use [2,3]. In many tropical regions, conversions of forests to agricultural land are ongoing [4,5], resulting in large-scale alterations of the water cycle and transpiration as a key flux. To measure, model and understand the effects of altered transpiration on the hydrological cycle, measurements

at the plant and leaf scale with sap flux probes and porometers are frequently applied [3,6,7]. While these hydrometric methods are commonly implemented at the leaf, plant or plot level, measuring transpiration at larger scales remains a challenging task [2,8,9]. Remote sensing techniques are often considered to be more cost-effective and labor-efficient than ground-based approaches, particularly for applications in agricultural and forest landscapes [10]. Remote sensing data at high temporal or spatial resolution, e.g., from satellites, offer opportunities for the extrapolation of point measurements but are associated with large uncertainties, especially for diverse mixed stands and agricultural areas [11]. Recently emerging drone-based remote sensing systems are promising to bridge the gap between leaf and plant scale measurement methods and catchment or landscape scale schemes [12]. Drones can operate close to the surface enabling a delimitation of single plant canopies but can also cover considerable areas with a single flight [13,14]. They can be equipped with a growing variety of light-weight sensors for diverse spectral ranges or structural measurements [15–17]. Image-based remote sensing approaches are limited in that only the top layer of the canopy is recorded and that measurement inaccuracies generally increase with increasing distance to the object of interest. Index approaches based on thermal remote sensing data such as the crop water stress index (CWSI) have been applied in semi-arid and arid areas using drone-recorded thermal data [18,19]. The application of the CWSI is most useful if a scarcity of water causes water stress in the plants [18]. However, if plants are well watered the explanatory power of such an index is reduced. Since leaf surface temperatures are highly variable during the course of the day, depending e.g., on solar irradiance [20] diurnal temperature patterns can be captured and evapotranspiration patterns can be calculated. To some extent, this might also be possible using other non-temperature-based approaches such as the normalized difference vegetation index (NDVI) that can capture the photosynthetic activity during the day [21]. For evapotranspiration, modelled results based on land surface temperature data often follow a linear relationship with ground-based measurements from eddy covariance systems [16,22]. For complex non-linear structures and relationships, e.g., between evapotranspiration and its controlling factors, statistical regression models can be supplemented with machine learning (ML) algorithms [23,24]. ML models are not explicitly programmed to represent biological processes but are data driven models that use a training data set and can later be applied to previously unknown data [10]. Among others, ML algorithms such as support vector machines (SVM), random forests (RF) and artificial neural networks (ANN) have previously been successfully applied to predict evapotranspiration or structural vegetation characteristics in a wide range of ecosystems [23,25–28]. In the SVM algorithm approach a regression plane and two parallel margins are fitted to the data in order to include as many instances as possible between the two margins and thus creating an explanatory model [29,30]. SVM regressors were previously successfully applied to regression problems with spatial data [28,31,32]. RFs are an ensemble learning approach where multiple trained decision trees are combined to provide an improved prediction performance [33,34]. Different designs of RF algorithms have previously been used to successfully predict e.g., crop water stress, evapotranspiration, above ground-biomass and basal area [10,26–28,35]. ANNs consist of multiple neurons that are combined in a set of layers [36] and are frequently used in many different environments [24–26]. Despite successful applications of ML algorithms to predict hydrological fluxes in the biosphere, the reliable quantification of the non-linear processes that govern water fluxes remains a challenging task [23]. Prerequisites for successful quantification include choice of an appropriate algorithm, a set of representative prediction variables and a sufficiently large data set [37].

The objectives of our study were (1) to compare linear statistical and ML approaches to calibrate models to predict sap flux and stomatal conductance measurements from drone remote sensing data and meteorological measurements (2) to identify the most important prediction variables for these models and (3) to compare the direct measurement methods (sap flux, porometry) with the modelling and drone-based methods for bias and interchangeability.

## 2. Methods

### 2.1. Study Site

The study was conducted in the lowlands of Sumatra, in Jambi province, Indonesia. Average elevation of the area is 47 m a.s.l., mean annual precipitation is 2235 mm·year$^{-1}$ and average annual temperature 26.7 °C [38]. The sites were situated in an oil palm plantation of the company PT Humusindo in a conventional monocultural oil palm plantation (Figure 1) and oil palm agroforests that were established in the context of a biodiversity enrichment experiment (EFForts-BEE, 103.2536 E, –1.9463 N) [39]. At the experimental agroforest sites and three years prior to this study, 40% of the oil palms were cut and native tree species were planted. In the conventional oil palm monoculture, palms were between 9 and 15 years old and the stem density was approximately 140 palms per hectare. At the time of our study in October 2016, average tree height was 4.7 m and average oil palm meristem height was 6.8 m. The six planted tree species differed in their habitat preferences including early successional species, as well as mid-to late successional species from old-growth (swamp) lowland forests. Therein, the three best performing species in terms of survival and growth (Peronema canescens, Archidendron pauciflorum and Parkia speciosa) are classified as early successional species and the species that experienced higher mortality (Shorea leprosula and Dyera polyphylla) are more closely associated with old-growth forest [40]. Oil palm is regarded as a pioneer species [41] but ecophysiological studies for palm species are generally rare.

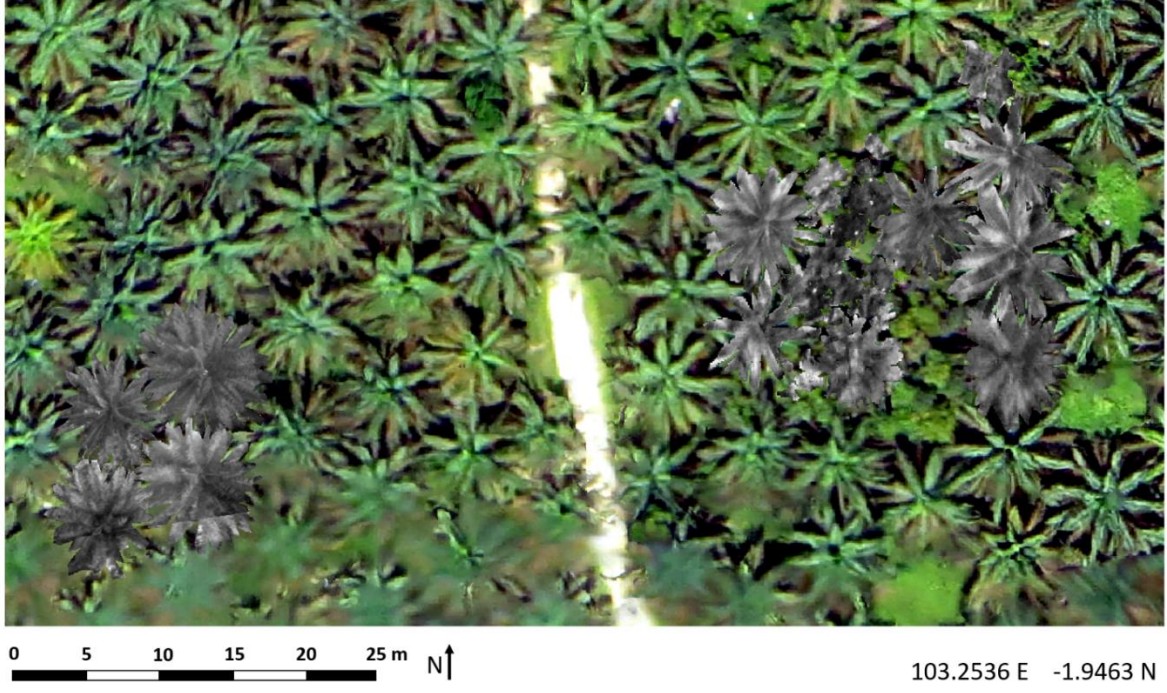

**Figure 1.** The studied monoculture oil palm plantation on the left-hand side and an exemplary oil palm agroforest site on the right side. Canopies of the measured trees and palms are shown with greyscale thermal images. Canopies were marked with 8-bit barcodes (small white dots in the image) to be recognizable from the air. The map shows an exemplary selection of all sampled canopies.

### 2.2. Data Acquisition

Two groups of data were acquired: the target variables, i.e., point measurements of sap flux and stomatal conductance with in-situ sensor applications, and a set of prediction variables recorded with a drone and a meteorological station.

### 2.2.1. Sap Flux Measurements

Eight oil palms (*Elaeis guineensis*) and 16 trees of the four most strongly represented native species, *Archidendron pauciflorum*, *Parkia speciosa*, *Peronema canescens* and *Shorea leprosula,* were equipped with sap flux sensors (Table 1). Four palms were located in a conventional monoculture area and four palms and all 16 trees in the agroforest sites. Oil palm sap flux was assessed with thermal dissipation probes [42] installed in the leaf petioles as described in [43]; sap flux density was calculated using calibrated, oil palm specific parameters [43] and then converted to oil-palm water use (mm·h$^{-1}$). For the dicot trees, we used the heat ratio method [44]; one sensor per tree was installed radially into the xylem at breast height (135 cm above the ground). For oil palm sapwood area was estimated following the methodology described in [43] and since the trees were still very small, the whole stem area was considered as water conducting area. Sap flux was then calculated using Sap Flow Tool version 1.4.1 (ICT International, Australia) and converted to tree water use (mm·h$^{-1}$). Further details on the applied sap flux methods are provided in [6]. Sap flux measurements were recorded simultaneously for all trees every 10 min over the course of two weeks. We used barcode markers on the sample trees and palms to facilitate their identification in the aerial images.

**Table 1.** Tree and palm details.

| Species | No. of Replicates | DBH * (cm) | Tree/Meristem Height (m) | Crown Projection Area (m$^2$) |
|---|---|---|---|---|
| *Archidendron pauciflorum, Fabaceae* | 4 | 7.6–11.0 | 6.5–8.8 | 9.8–17.6 |
| *Parkia Speciosa, Fabaceae* | 4 | 5.5–9.5 | 6.0–9.0 | 4.5–13.0 |
| *Peronema canescens, Lamiaceae* | 4 | 8.0–11.0 | 6.0–8.6 | 7.8–11.0 |
| *Shorea leprosula, Dipterocarpaceae* | 4 | 4.2–5.8 | 3.3–5.2 | 1.1–3.9 |
| *Elaeis guineensis, Arecaceae* | 8 | – | 5.19–7.11 | 64.0–103.7 |

* DBH refers to measurements of the diameter at breast height (135 cm above the ground).

### 2.2.2. Stomatal Conductance Measurements

Stomatal conductance measurements were conducted using three porometers (AP 4, Delta-T Devices Ltd, Burwell, Cambridge, UK). The sunlit areas of the canopies were reached using scaffoldings. Three palms were assessed using nine fronts per palm as described in [7]. For trees, we successfully measured three *Archidendron pauciflorum*, three *Peronema canescens* and four *Shorea leprosula (*Table 1). The species *Parkia speciosa* is not included in the stomatal conductance dataset because leaflets were too small to fully cover the porometer measurement chamber. Seven to ten sunlit leaves each were measured on three branches per tree and mean values were calculated.

The measurement chamber was placed only on the abaxial epidermis of the leaves, assuming that the majority of stomata of almost all species in tropical forests is located on the abaxial epidermis [45,46]. This approach further supported the use of the built-in light sensor in the porometers sensor head. Each sample palm or tree was measured between three and five times per day, starting from 9 a.m. to 4 p.m. local time over the course of two weeks. This variation in measurement times was mainly caused by differences in relative humidity which affected the time period to complete the measurement cycle of the porometer devices.

### 2.2.3. Drone-Based Image Acquisition

We used a multicopter drone (MK EASY Okto V3; HiSystems, Moormerland, Germany) equipped with a thermal and an RGB camera mounted in a stereo setup on a gimbal set to nadir perspective. The radiometric thermal camera was a FLIR Tau 2 640 (FLIR Systems Inc., Wilsonville, OR, USA) attached to a TeAx Thermo-capture module (TeAx Technology GmbH, Wilnsdorf, Germany). The sensor covers spectral bands ranging from 7.5 to 13.5 µm with a relative thermal accuracy of 0.04 K and

an absolute thermal accuracy of 1–2 K. Since we used a radiometric thermal camera to conduct our canopy surface temperature measurements, the data could directly be exported for analysis from the camera without further need to translate greyscale images to the corresponding temperatures. The RGB camera was a Sony A5000 (Sony Corporation, Tokyo, Japan) with an EPZ 16–55mm F3.5-5.6 OSS lens that was set to 16 mm focal length. During an 11-day campaign in October 2016 a total of 103 flights were conducted to record canopy surface temperatures simultaneously to the sap flux and stomatal conductance measurements. RGB images were merged using Photoscan 1.3.0 (Agisoft LLC, St. Petersburg, Russia) to create a geo-referenced orthomosaic map. All thermal infrared (TIR) images were then referenced to the RGB map, the canopies were delineated and single palm and tree canopies were extracted using QGIS3 version 3.6 Noosa (QGIS Development Team, 2020).

### 2.2.4. Meteorological Measurements

Meteorological measurements were conducted at a station located in the studied oil palm plantation. The station was equipped with a global radiation sensor (CMP3 Pyranometer, Kipp & Zonen, Delft, The Netherlands), two thermohygrometers (type 1.1025.55.000, Thies Clima, Göttingen, Germany), a net radiometer (NR Lite2, Kipp & Zonen) and a 3-cup anemometer and a wind direction sensor (both Thies Clima). Data were stored on a logger every 10 min (see [47] for details).

### 2.2.5. Data Pre-Processing

Single canopies were marked with 8-bit barcodes that were visible in the RGB, as well as the thermal spectrum, and canopy surface temperatures were cut from the thermal images for each canopy and flight. We used the QGIS3-plugin QWaterModel [48] to analyze the canopy surface temperatures by extracting the key metrics mean, median, standard deviation, coefficient of variation, kurtosis and FPCS (Fisher-Pearson coefficient of skewness). We then applied the energy balance model DATTUTDUT [49] based on the canopy temperatures, again using the QGIS3-plugin QWaterModel, and extracted the same key metrics for the resulting fluxes. To calculate the fluxes using the DATTUTDUT model we applied both a fully modelled net radiation approach and a short-wave irradiance-based estimation approach [22]. We used the *pandas* library [50] in Python 3.6.9 to merge the datasets according to the individual plant and recording time. We set the maximum time delta to an hour and matched 1710 datasets for sap flux and 877 for stomatal conductance. The average time offsets for sap flux and stomatal conductance with the remote sensing data were 70 and 1481 s, respectively. The final data set contained 96 variables including the two target variables (sap flux and stomatal conductance), micrometeorological variables (short-wave irradiance, barometric pressure, air temperature, wind speed and direction, relative humidity and the vapor pressure deficit (VPD)), duplications in measurement efforts using several measurement devices and a multitude of variables from drone recorded data and modelled products such as land surface temperatures, the leaf based VPD, evaporative fraction and different evapotranspiration estimates (incl. various dispersion metrics such as mean, median, standard deviation, coefficients of variance, kurtosis and the FPCS). Furthermore, the data set contained local time encoded as cyclical feature (with sinus and cosine as a variable), the canopy area, the number of pixels and atmospheric emissivity and transmissivity. All input variables and their abbreviations and units are shown in Table 2.

**Table 2.** Input variables, abbreviations and units.

| Abbreviation | Description | Unit |
|---|---|---|
| Meteorological data | | |
| Short wave irradiance *** | measured short-wave irradiance | $(W \cdot m^{-2})$ |
| Air temperature *** | measured air temperature | $(^{\circ}C)$ |
| Barom. pressure | measured barometric pressure | (hPa) |
| Wind speed | measured wind speed | $(m \cdot s^{-1})$ |
| Wind direction | measured wind direction | $(^{\circ})$ |
| Relative humidity | measured relative humidity | (%) |
| VPD gen | vapor pressure deficit based on air temperature | (kPa) |
| Drone-images/thermal-data | | |
| Canopy area | canopy area derived from aerial image | $(m^2)$ |
| Number of pixels | sum of canopy area pixels | (-) |
| LST * | land surface temperatures | (K) |
| VPD leaf | vapor pressure deficit based on land surface temperatures | (kPa) |
| Model results ** | | |
| Rn * | net radiation from model output | $(W \cdot m^{-2})$ |
| LE * | latent heat flux from model output | $(W \cdot m^{-2})$ |
| H * | sensible heat flux from model output | $(W \cdot m^{-2})$ |
| G * | ground heat flux from model output | $(W \cdot m^{-2})$ |
| EF * | evaporative fraction from model output | $(W \cdot m^{-2})$ |
| ET * | evapotranspiration from model output | $(W \cdot m^{-2})$ |
| atmos. transmissivity | atmospheric transmissivity from model | (-) |
| atmos. emissivity | atmospheric emissivity from model | (-) |
| Other | | |
| local time sinus | cyclic local time variable | $(^{\circ})$ |
| local time cosinus | cyclic local time variable | $(^{\circ})$ |

* Data variables used in several metrics: mean, median, std. dev (standard deviation), coef. var. (coefficient of variation), kurtosis, fpcs (Fisher-Pearson coefficient of skewness). ** Model results from two different applications of the DATTUTDUT model [49]: DM is the original version of the model using a fully modelled net radiation, DS is a model version that uses net radiation from a partly measured (measured short-wave irradiance) and otherwise modelled approach. *** Measurements of these variables are duplicated. This is indicated by the numbers of measurement devices (1,2).

## 2.3. Prediction Models

The data sets for sap flux prediction (n = 1710) and stomatal conductance prediction (n = 877) were each split into a training set and a test set (70% and 30% of the data, respectively) using scikit learn [51] and a pseudo-random seed of 42 to guarantee reproducibility [52]. Further data sets for each species and a joined data set for all trees were created in a similar way. We performed a multicollinearity test based on a variance inflation factor with Pearson's r (cut-off value > 0.9) and backward elimination of variables based on the Akaike information criterion (significance level $p < 0.05$) and removed variables accordingly reducing the number of input variables from 96 to 42. We decided to use a multiple linear regression (MLR) as our baseline method as it represents a well-known standard approach in statistics. We further applied support vector machine regressors (SVM), random forest regressors (RF) and artificial neural network regressors (ANN) to build prediction models.

### 2.3.1. Multiple Linear Regression

Multiple linear regression (MLR) is an approach to model the relationship between the explanatory input variables and a response variable by fitting a linear equation expressed as a regression plane [36]. In our study, we used a least-squares model that minimizes the sum of squared vertical deviations from each point to this regression plane [53]. MLR was computed using the *LinearRegression* regression

class from the *Scikit-learn* package in Python [51]. MLRs were previously successfully used to predict daily transpiration, evapotranspiration and basal area of trees from spatial data [24,27,31].

### 2.3.2. Support Vector Machine

Support vector machines (SVMs) model the relationship between explanatory variables and the response variable by fitting a regression plane with parallel margins to the data in order to include as many instances as possible between the two margins [29,30]. This regression plane is also referred to as a hyperplane if several explanatory variables are used [54]. SVMs are mostly known as classifier algorithms, but can also be applied for regression problems [55]. We used the SVR method of *Scikit-learn* package and a linear kernel to build SVMs [51].

### 2.3.3. Random Forest

Decision trees are predictive models that use recursive partitioning for classification or regression tasks [56]. Random forests combine the results of a set of individually trained decision trees [33]. This principle is called ensemble learning and has shown to improve the predictive performance [34,54]. Two widely used methods of ensemble learning are bootstrap aggregation referred to as *bagging* [57] and *boosting* [58,59].

In the bagging method multiple replicates of the original learning data set are created by bootstrapping with replacement [60]. The decision trees are then trained with these different variations of the original data set and the results of the individual decision trees are averaged [60]. Bagging reduces the variance of simple models and helps to avoid overfitting of more complex models [61]. We trained 2000 trees and used the *RandomForestRegressor* method of the *Scikit-learn* package [51] to build our model. The idea behind boosting is to combine a set of weak and moderately inaccurate decision trees and average their predictions to create very accurate predictions [62]. For this approach we trained 4000 trees and used the adaptive boosting algorithm (*AdaBoost*) introduced by [35] from the Scikit-learn package [51].

### 2.3.4. Artificial Neural Network

Artificial neural networks (ANNs) are inspired by biological brains and consist of multiple neurons that are organized in a set of layers [36]. ANNs are ideal to identify complex non-linear relationships between in- and output data sets and particularly useful in regression problems with processes that are difficult to capture entirely [24,25]. We used the *Sequential*, *Dense* and *KerasRegressor* methods of the *keras* framework [63] to build an ANN with the typical multiple perceptron type (MLP) architecture which was recently used to predict tree metrics from spatial data [31,64]. Similar ANN designs have been used to estimate evapotranspiration and transpiration in a wide range of ecosystems [24–26]. We used rectified linear units (ReLU) for the activation function in the input and the three hidden layers and a linear activation function for the output layer.

### 2.3.5. Variable Importance

To estimate the importance of each predictor variable for the regression model, we decided against a removal-based approach as described in e.g., [31] and opted for a randomization-based permutation test (also called mean decrease accuracy) using the *PermutationImportance* method from the *eli5* package [33,65]. Hereby, the values of a single input variable of the prediction dataset are randomized (and not left out) and its effect on the prediction accuracy of the regression model is measured [65].

### 2.3.6. Statistical Analyses of Predicted vs. Measured Values

To evaluate the prediction models, we compared measured and predicted values from the test data set. We calculated model accuracy in % (defined as 100 subtracted by the mean absolute percentage

error (MAPE)) as a general prediction performance indicator. We limited the model accuracy to a range of 0%–100 %, setting potential negative values to zero if the prediction error MAPE becomes larger than 100%. The closer this indicator approaches 100% the closer the model predictions are to the actual observations. We further calculated the mean absolute error (MAE) that measures the average magnitude of errors between the prediction and the test data set and is displayed in percentage using the mean of the observation data set as a base. Further the root mean squared error (RMSE) that indicates the square root of the squared errors between prediction and test data set was calculated using the same base for percentage transformation than the MAE. Both the MAE and RMSE are indicators that measure the average prediction error, however the RMSE penalizes more on extreme errors. The MAE focusses on average errors alone and is therefore a more general indicator. Both the MAE and the RMSE are negative indicators, the higher the indicator value the lower the precision of the predictions. We further calculated the coefficient of determination ($R^2$) to indicate how well the resulting model fits the data. Variable importance of each predictor variable was assessed using a permutation test [65]. Single variable importance was averaged over all data sets for each sap flux and stomatal conductance and might vary for single species. To compare the prediction methods, we used a non-parametric Passing-Bablok regression [66–68]. The python *MethComp* package [69] was used for the computation of the Passing-Bablok regressions. The Passing-Bablok regression outputs a regression line of which the confidence intervals of slope and intercept are especially interesting for a comparison of the different methods. If the confidence intervals of the slope and the intercept include 1 and 0, respectively, there is no statistically significant bias between the methods [53,68]. Linearity of the data is a crucial assumption for Passing-Bablok regressions [68]; and was checked visually. All statistical analyses were computed with Python 3.6.9 using *pandas* [50], *NumPy* [70,71], *SciPy* [72], *statsmodels*, *scikit learn* [51], *keras* and *eli5* packages and libraries. Graphs and figures were created with *Matplotlib* [73] and *seaborn* [74] libraries.

## 3. Results

Data was acquired on 16 days, between the 1st and 16th of October 2016, from 9 a.m. to 4 p.m. local time. During the field measurements air temperature ranged from 25.2 to 36 °C, incoming short-wave irradiance ranged from 143 to 1124 W·m$^{-2}$ and relative humidity ranged from 49.6% to 94.4%. Average wind speed was 1.26 m·s$^{-1}$ predominately coming from South-East. Meteorological conditions were therefore highly variable and allow for training the algorithms across multiple weather conditions. Sap flux for both oil palm and trees showed strong diurnal patterns, with near-zero values at night time and often near-noon maxima (following the diurnal patterns of integrated daily radiation Rg and vapor pressure deficit VPD). Daytime maxima for sap flux ranged from 0.831 mm·h$^{-1}$ (*Parkia speciosa*) to 2.544 mm·h$^{-1}$ (*Archidendron pauciflorum*) daytime maxima for stomatal conductance ranged from 2153.7 mol·m$^{-2}$·h$^{-1}$(*Shorea leprosula*) to 11813.4 mol·m$^{-2}$·h$^{-1}$(*Peronema canescens*).

### 3.1. Prediction Performance

Highest model accuracy was achieved by random forest models (both bagging and boosting) predicting sap flux for individual tree species, especially *Archidendron pauciflorum* where a model accuracy of well over 90% was reached (Figure 2). Across all tree species, accuracy was 60% and 52% for oil palm (RF bagging). Prediction of sap flux across all canopies, including dicot trees and palms, was unsuccessful with all applied algorithms. For predictions of stomatal conductance, RF bagging was again the method with the highest model accuracy, consistently yielding prediction accuracies of 60% for the tree species and 48% for oil palm (Figure 2). For stomatal conductance, predictions across a data set comprising all canopies were successful, with 60% accuracy for RF bagging.

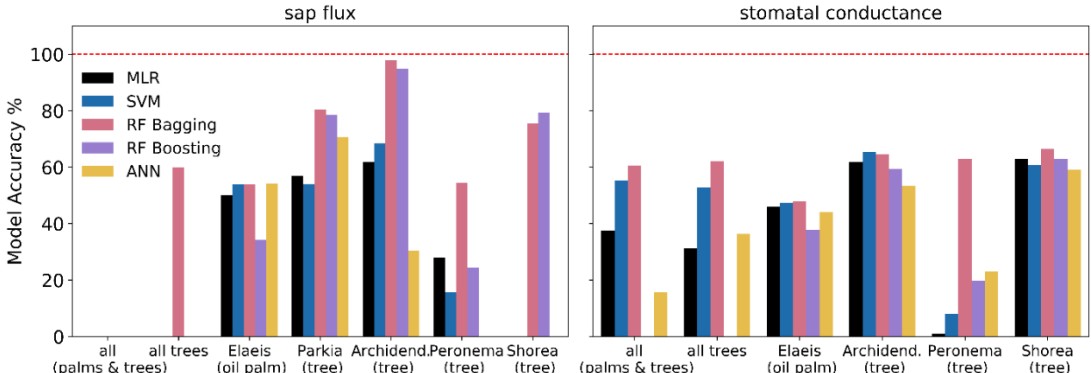

**Figure 2.** Model accuracy % comparing measured and predicted values of sap flux and stomatal conductance. Abbreviations: MLR—multiple linear regression; SVM—support vector machine; RF—random forest; ANN—artificial neural network.

A comparison of error metrics (MAE and RMSE) showed that errors for the random forest algorithms and particularly RF bagging were comparatively low for both sap flux and stomatal conductance prediction (Figures 3 and 4). The simpler algorithms such as MLR and SVM but also the more complex ANN frequently produced higher errors. Generally, the errors for stomatal conductance predictions are more evenly distributed across algorithms and target groups than for sap flux prediction. For instance, MAE and RMSE for *Shorea leprosula* were relatively small and homogeneous across algorithms for stomatal conductance, but ranged from under 30% (RF bagging) to over 100% (ANN) for sap flux.

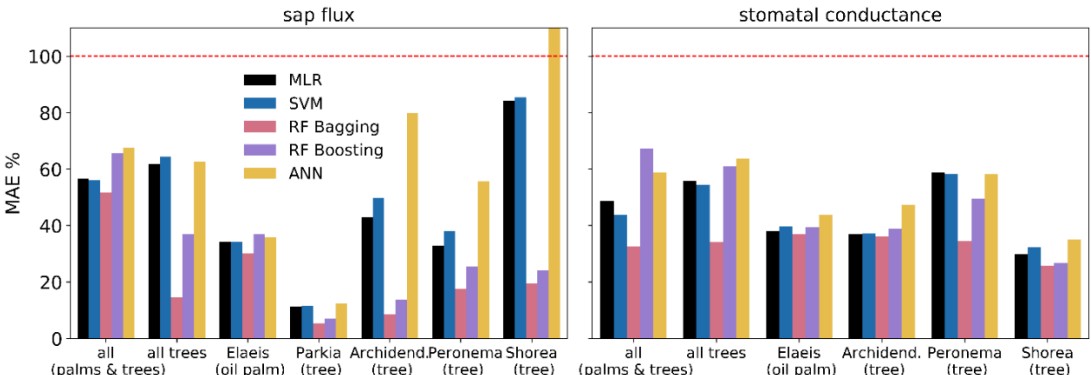

**Figure 3.** Mean absolute error (MAE) of predicted and measured values for sap flux and stomatal conductance.

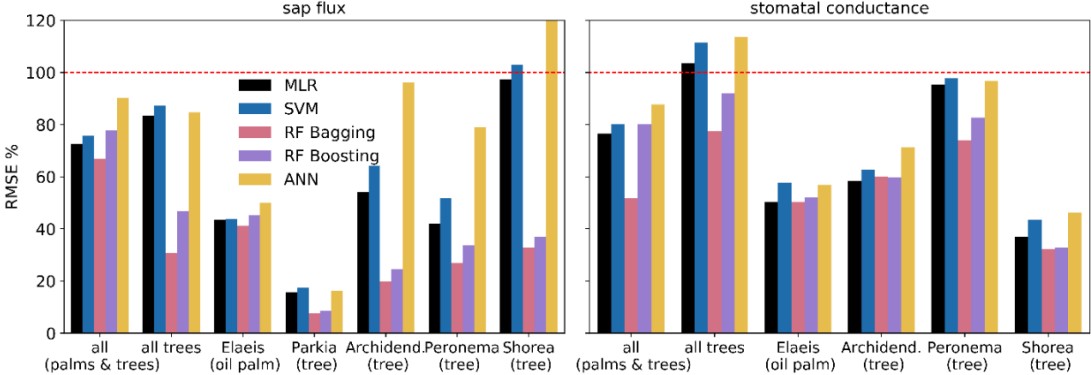

**Figure 4.** Root mean square error (RMSE) of predicted and measured values for sap flux and stomatal conductance.

Both RF algorithms resulted in predictions highly congruent with the sap flux measurements in terms of $R^2$ (Figure 5); across all tree species, as well as for each individual tree species, $R^2$s were close to or higher than 0.8 (RF bagging), while they were close to 0.6 for oil palm. The predictions for stomatal conductance are generally less congruent with the measurements than for sap flux. However, the RF bagging algorithm does achieve $R^2$s around 0.5 or higher across all canopies and across all tree species, as well as individually for two of the four species.

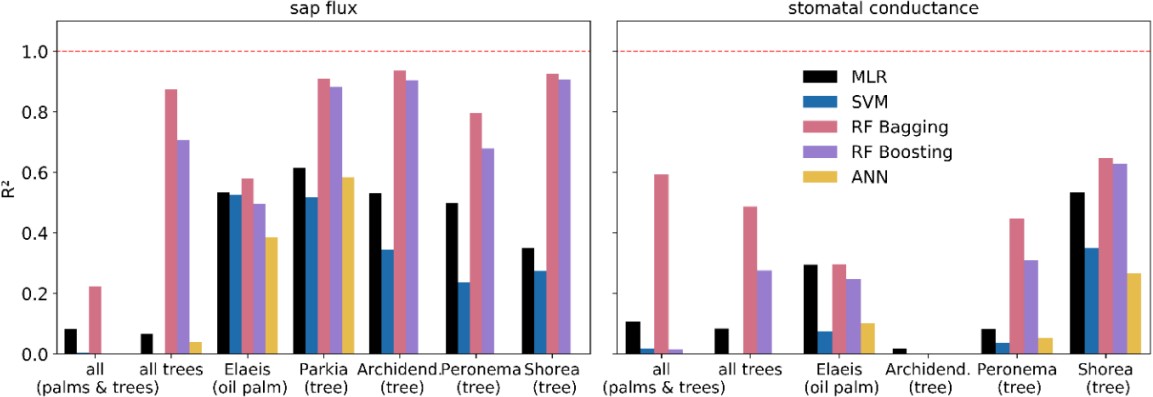

**Figure 5.** Coefficient of determination ($R^2$) for predicted and measured values for sap flux and stomatal conductance.

## 3.2. Method Comparison

Predicted values for sap flux from the RF algorithms almost always showed a linear relationship with their measured counterparts, whereas this can only be observed for the MLR and SVM algorithms for oil palm datasets and was never detected for the ANN algorithm (Figure 6). There was no linear relationship of measured and predicted values for stomatal conductance for all prediction algorithms, the results are therefore not included into a figure. Sap flux predictions for all trees and oil palm (but not all canopies), as well as for *Archidendron pauciflorum* showed no significant continuous or systematic bias from the measurements when the RF bagging algorithm was applied (Figures 6 and 7), which indicates interchangeability of the methods.

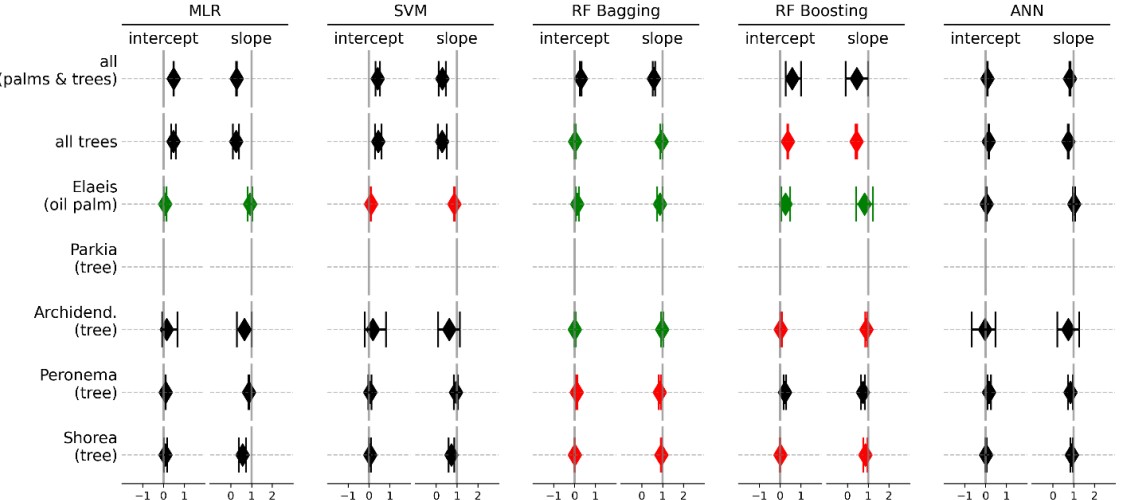

**Figure 6.** The 99% confidence intervals of slope and intercept from model II Passing-Bablok regression for sap flux. The results in green show no significant bias between measured and predicted values. The results indicated in red fulfilled the assumptions of linearity but a significant difference between measured and predicted values was found. The results displayed in black did not meet the assumption of linearity and are therefore not to be considered.

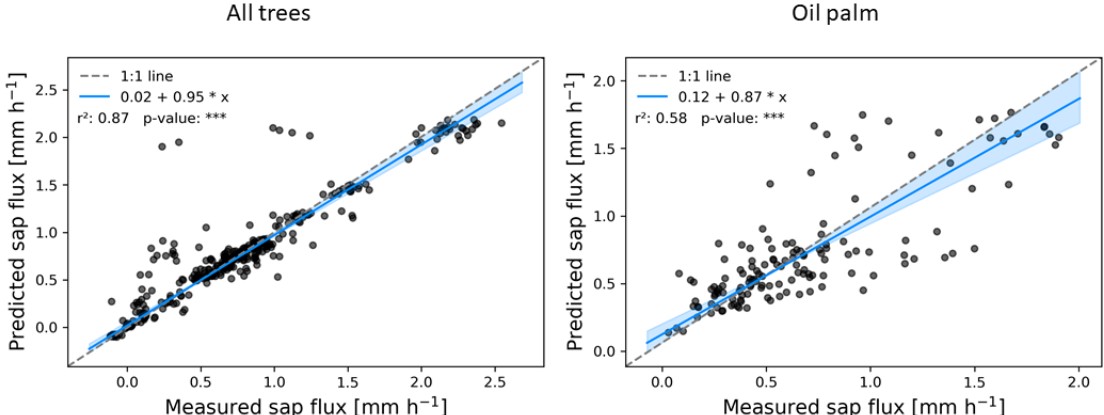

**Figure 7.** Measured and predicted sap flux for all tree and palm canopies from the RF bagging algorithm. The regression line of the Passing-Bablok regression is displayed in blue. The coefficient of determination is indicated by $r^2$ and the *** indicate a *p*-value well below 0.001.

Except for some outliers, the measured and RF bagging predicted values for sap flux in trees closely follow the 1:1 line, while showing more variance for oil palm (Figure 7). The same was observed for oil palm canopies where the MLR and RF boosting algorithms predicted sap flux without significant bias or errors (Figure 6). For species specific predictions, linearity was found for both RF algorithms, but except for *Archidendron pauciflorum* a significant systematic and continuous bias was detected. None of the algorithms showed potential in predicting sap flux of *Parkia speciosa*.

### 3.3. Variable Importance

A total of 96 input variables (or features) were available for our analysis. After applying a multicollinearity test and a backward elimination, the remaining 42 variables were used to train the prediction algorithms. To improve the distribution of measurement efforts and the relevance of input features, we performed a permutation importance analysis. Therein, the numbers of most important variables that explain 95% of the model outcome are highly variable (Figure 8). For the MLR, a very low number of only up to seven input variables (see Figure 9 for details) are required to explain most of the model prediction results for both sap flux and stomatal conductance. For the SVM algorithm, between 10 and 20 variables explain 95% of the model outcome; a smaller number of variables explain the sap flux results, while the number of variables was higher for stomatal conductance predictions. The RF algorithms showed the highest variations ranging from only one variable up to 20 for sap flux and generally using 15 to 20 main variables for stomatal conductance to explain most of the model's outcome. The ANN algorithm uses an intermediate number of around 9 to 17 important variables for both sap flux and stomatal conductance prediction.

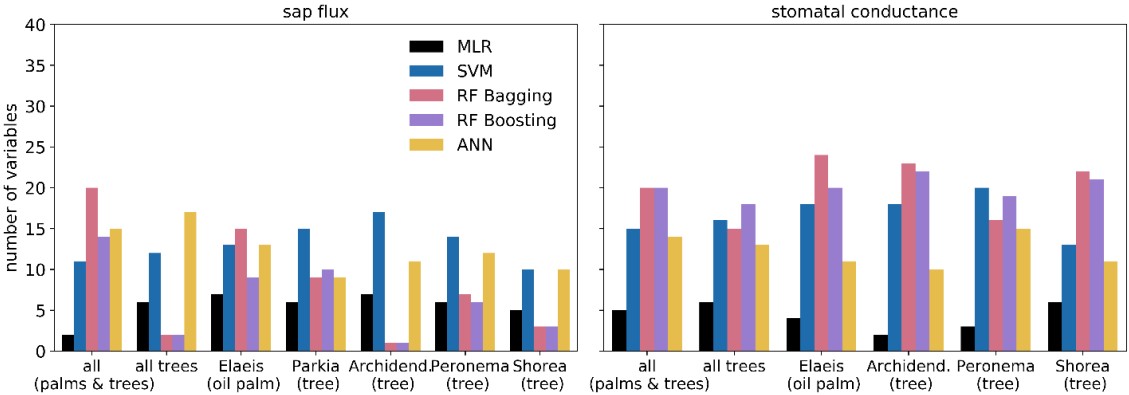

**Figure 8.** Number of most important input variables that explain 95% of the corresponding model.

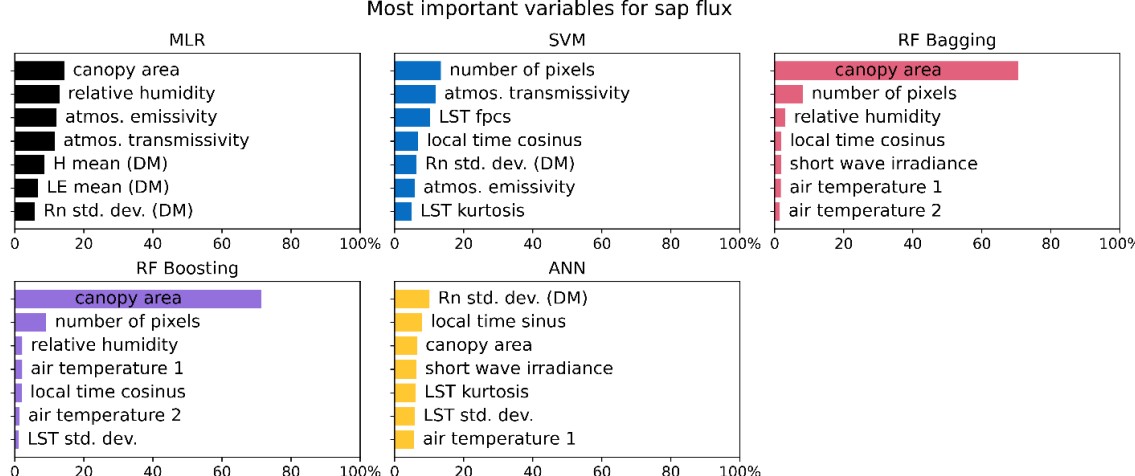

**Figure 9.** The seven most important input variables averaged over all data sets (different canopies and combinations) for the corresponding models when predicting sap flux.

Variable importance (expressed in %) averaged for sap flux and stomatal conductance over all species shows big differences for the MLR algorithm, where variable importance for sap flux is very homogeneous and the prediction of stomatal conductance is dominated by the latent heat flux as derived from the DATTUTDUT model (Figures 9 and 10). A similar but less pronounced pattern can be observed for the SVM regressor where the main prediction variables for sap flux are very homogeneous and for the prediction of stomatal conductance barometric pressure is the prevalent variable.

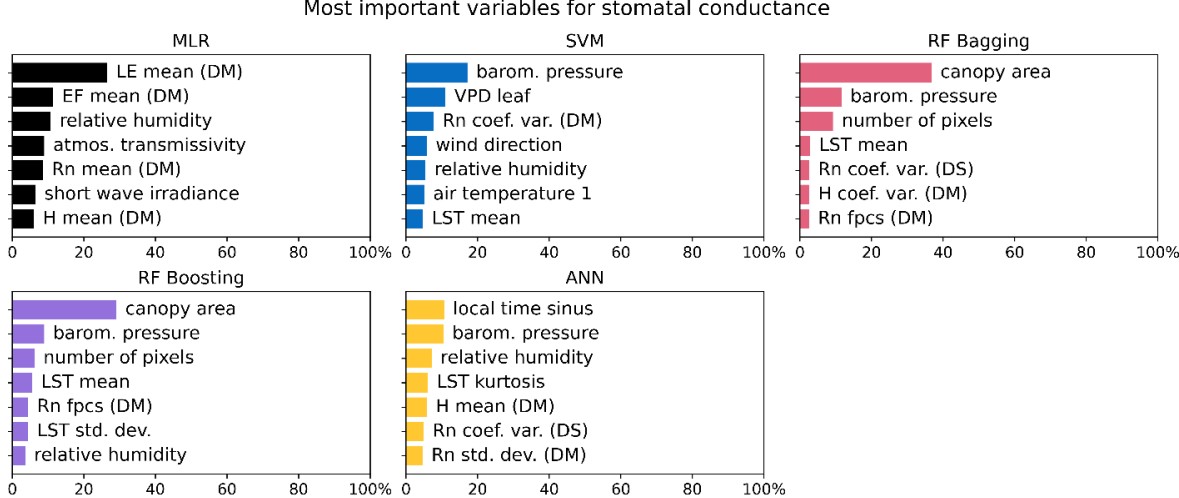

**Figure 10.** The seven most important input variables averaged over all data sets (different canopies and combinations) for the corresponding models when predicting stomatal conductance.

Both RF algorithms mainly rely on the canopy area as a main prediction variable for both sap flux and stomatal conductance. Another observation is the second variable that is of a much less pronounced importance for the model results: both RF algorithms use the number of pixels for sap flux predictions and the barometric pressure for stomatal conductance predictions (Figures 9 and 10). The ANN algorithm shows a very homogeneous distribution with no prevalent variable for both sap flux and stomatal conductance prediction (Figures 9 and 10).

## 4. Discussion

This study analyzes the applicability of machine learning approaches for ecological datasets, in our case thermal remote sensing data from drones and micrometeorological data to predict sap flux and stomatal conductance as two key processes related to the water balance of individual plants and ecosystems. Model accuracy was generally higher and errors were lower for sap flux than for the stomatal conductance predictions. Therein, the applied prediction algorithms showed substantial differences regarding input requirements and variable importance. The prediction results from the RF bagging method were found to be interchangeable with ground-based measurements for some canopy types and species.

*4.1. Prediction Performance*

In our study MLR representing the simplest algorithm produced results of intermediate quality often with lower errors than the more complex SVM algorithm. This stands in contrast to previous studies, where the SVM algorithm produced more congruent estimates, e.g., sap flux from a range of directly measured variables or biomass from remote sensing data [24,28]. Since the creation of a representative model with MLR requires a prevalent linear relationship of at least some input variables with a given target variable, we assume that given a supplementary set of ground-based measurements MLR can still be a simple alternative to more complex machine learning algorithms. In contrast to a previous study where evapotranspiration was successfully predicted from hydro-climatic variables with the SVM algorithm [55], this algorithm did not produce satisfying results in our study. On average, model accuracy was higher for the SVM when predicting stomatal conductance than when predicting sap flux, but coefficients of determination indicate that neither model predicts the target variable accurately. Since we used a linear kernel, the performance of the SVM algorithm highly depends on the linear relationship of at least some input variables with the according target variable.

Overall, both RF algorithms (bagging and boosting) resulted in similar, high quality predictions, with the bagging algorithm performing slightly better. Previous studies predicting water stress from remote sensing data showed similar results, i.e., the bootstrapping-based bagging option slightly outperforming the boosting algorithm but with small overall differences [75]. Compared to the other applied prediction methods, both RF algorithms showed high model accuracy and low errors with a high congruence of measured and predicted values, particularly for sap flux. Likewise, RF algorithms outperformed several other algorithms in studies where above-ground biomass was estimated from remote sensing data [28,31]. A previous study applying similar feed-forward ANN algorithms but using ground-based input data showed satisfactory results for sap flux prediction [76]. While we expected better results from the ANN algorithm due to its complexity, the parameterization of the method is known to be difficult and highly sensitive to variations in input parameters [31,77].

*4.2. Method Comparison*

Using RF algorithms, sap flux can be predicted from remote sensing data in close congruence to measurements without producing many outliers (Figure 8). While analyzing tree and palm canopies separately, sap flux can well be predicted by the RF bagging algorithm, while for the whole dataset including both, trees and palms, sap flux was not adequately predicted. The reasons are likely underlying differences in physiology between dicot trees and monocot palms; this e.g., includes size and distribution of water-conductive vessels in the stem and crown and leaf architecture. As such, previous ecohydrological assessments in the study region pointed to vast differences in water use between oil palms and trees including rubber trees of similar age, with substantially higher per-tree and stand transpiration rates of oil palms [78,79]. Drone-derived crown metrics of oil palms and adjacent trees further suggested that oil palms transpire two-times more water per unit of crown volume as agroforest trees [6] and about five-times more per unit of crown surface area than rainforest trees [80], which may well be the reason why the joined analysis of trees and palms was unsuccessful.

A further reason for the difficulties in predicting sap flux across tree and oil palm canopies might be of methodological nature, as palm sap flux was assessed in leaf petioles with thermal dissipation probes [42] and tree sap flux was assessed in the lower stems with the heat ratio method [44]. The good prediction performance of RF algorithms compared to e.g., ANN algorithms was previously also described for predicting potential evapotranspiration [26]. RF was further found to be the best of several algorithms for aboveground biomass estimation from remote sensing data [28].

The rather complex ANN algorithm showed no convincing results in our study. Potential reasons include that variables might not have been captured due to a lack of a sufficing number of free weights [24], or that a surplus of free weights might have caused an over-fitting of the model which lacked the generalization to predict reasonable results from our test data set [81]. Even more than for the other algorithms, optimal layout of the ANN highly depends on the input variables [82].

None of the applied models could predict sap flux in *Parkia speciosa*, despite comparatively low errors and a promising model accuracy, as well as high congruence with measurements. Reviewing individual canopies of *Parkia speciosa* showed that, due to its pinnate leaves and open crown structure, canopies are partly barely visible in the thermal images and likely have a high contribution of pixels representing the soil or mixed soil-canopy pixels.

In our study, major challenges remain with the prediction of stomatal conductance. The data set for stomatal conductance differed strongly from the data set for sap flux. Average time offsets of directly and remotely measuring stomatal conductance were much bigger than for sap flux and measured and modelled input variables were not ideally suited for stomatal conductance prediction. The use of an additional set of multispectral images and resulting indices such as the NDVI (normalized difference vegetation index) or the EVI (enhanced vegetation index) could potentially enhance the prediction of stomatal conductance [83] especially from remote sensing data. The use of other TIR based indices such as the crop water stress index (CWSI) might be a useful addition to the training data set. TIR data might not be the ideal source for stomatal conductance predictions, since other studies were able to predict stomatal conductance from hyperspectral reflectance spectroscopy data using RF and ANN algorithms [84]. Further, the sample size for sap flux was much larger than for stomatal conductance due to a limited amount of porometry devices and the non-automated nature of the measurements. Furthermore, the stomatal conductance measurements represent only a small portion (centimeter scale) of a given canopy. The identification and subsequent upscaling from this small leaf area to a whole canopy thus likely introduces substantial errors into the predictions.

Despite the very encouraging results, the genericity of our study is limited as it encompassed only one study region (tropical lowland environment) and only oil palm systems therein and as measurements covered a period of less than one month. On the other hand, the lowland tropics have a large extent, oil palm continues to expand throughout the tropics and the equatorial climate is often quite stable throughout the year. Further studies are necessary to confirm whether the applied schemes can be applied across regions, vegetation types and seasons. A general drawback of machine learning models is that they usually lack causal relations that would enable a biological interpretation [31,85]. Our study is based on approaches that are often described as the traditional statistical (MLR) and traditional machine learning (SVM, RF, ANN) methods [23], while more complex approaches such as the stochastic gradient boosting, which combines the advantages of boosting and bagging [28], were not within the scope of this study. This leaves room for further expanding and refining the here presented approaches.

In summary, RF approaches worked best for predicting sap flux, while a model that can predict stomatal conductance without bias was not found.

### 4.3. Variable Importance Evaluation

The output quality of prediction algorithms highly depends on how well the input variables represent the ecosystem and the target variables [27]. For sap flux and stomatal conductance, one would generally expect a strong influence of environmental variables such as the vapor pressure deficit (VPD)

or solar radiation. The overall number of variables that explained 95% of our MLR model was low, never surpassing seven variables (Figure 8). This is a strong indicator for either no or non-linear relationships between most input variables and the output variables. The seven most important variables averaged over all input data set configurations (mixed and single species) of the MLR model in our study show that canopy area, as well as relative humidity have the greatest impact when predicting sap flux (Figure 8). These results only partly resemble variables from other studies where air temperature and VPD related variables such as relative humidity played a key role in MLR derived models to predict sap flux [76]. The most important variable for stomatal conductance predictions is the mean latent heat flux derived from the DATTUTDUT model (Figure 9). Surprisingly, wind speed was not amongst the most important input variables for the MLR algorithm whereas an influence of wind speed on both target variables was shown in previous studies (e.g., [86]). Asides from meteorological variables, soil moisture is considered to be a classic driver of sap flux and thus transpiration (e.g., [87,88]). While our data set did not comprise soil moisture data, previous assessments in the same study region showed no significant influence of the typically rather small soil moisture fluctuations on (evapo)transpiration of oil palms and trees [3,89,90], with exception of a strong El Niño event [90], which did not occur during our time of study. In contrast, soil moisture can be strongly limiting factor in (semi)arid regions, as was e.g., reported using a MLR approach [24]. We could not identify dominating input variables that would explain the model built by the SVM algorithm for sap flux. The prediction from the SVM algorithm for stomatal conductance was slightly dominated by the variable barometric pressure (Figure 10). The VPD for leaves based on the drone recorded leaf surface temperatures plays a further key role when predicting stomatal conductance with the SVM algorithm. However, none of the variables explained more than 20% of the final model. In contrast, a previous study successfully applied a SVM to predict potential evaporation using solar radiation, relative humidity, air temperature and wind speed as input variables [91]. The number of variables that explains 95% of the model results was very low for both RF algorithms when predicting sap flux. In contrast, the RF algorithms used among the highest numbers of variables when predicting stomatal conductance. For both, canopy area was the (clearly) dominating predictor. With simple linear regression analysis, no relationship between canopy size and sap flux could be found, mainly because the canopy size is a constant variable while sap flux quantity varies during the day. We assume a non-linear relationship between canopy size and sap flux, which requires auxiliary variables to be predicted. Remotely sensed canopy size is an important factor for plant transpiration and was already found to be a suitable predictor in previous studies [6,80]. A general advantage of RF algorithms is their ability to produce accurate predictions even from small samples and with a large set of independent input variables [31]; in our study, they performed better than all other applied algorithms.

The ANN algorithm mainly uses an intermediate number of input variables compared to the small set of MLR based models and the large sets of the SVM models. Compared to the other algorithms, the ANN does not show any clearly dominant input variable. An interesting observation is that the ANN algorithm uses cyclic local time variables among its most important input variables. This seems to be a logical approach as sap flux is known to follow a daily course with near-noon maxima and night time near-zero minima in the study region (e.g., [78,79]). One previous study found that the best results for stand transpiration prediction were achieved by using climate data, soil water content and canopy properties as key input variables [24], which are partly also represented in our ANN derived prediction models.

## 5. Conclusions

Drone remote sensing and in situ meteorological observations in conjunction with machine learning algorithms can be considered reliable methods for the prediction of sap flux. For tree and palm canopies, random forest regressors predicted interchangeable results without significant bias when compared to direct sap flux measurements. The prediction of stomatal conductance from remotely

sensed data was less successful and requires further research. Our study complements the asset of available sap flux approaches by a reliable method for drone-based sap flux prediction.

**Author Contributions:** The study was conceptualized by D.H. in cooperation with H. (drone and sap flux measurements) and B.S. (stomatal conductance measurements). F.E. led the writing of the paper with help from A.R. and D.H. supervised the work. F.E. collected and processed the drone data, J.A. and A.R. collected the sap flux data and P.-A.W. collected the stomatal conductance data. F.E. conducted data processing, model application, statistical analysis and production of plots in cooperation mainly with D.H. and A.R., F.E., D.H. and A.R. created a first version of the manuscript, which was further improved in a cooperation of all authors. All authors have read and agreed to the published version of the manuscript.

**Funding:** This study was funded by the Deutsche Forschungsgemeinschaft (DFG, German Research Foundation) project number 192626868—SFB 990 (subprojects A02 and B04) and the Ministry of Research, Technology and Higher Education (Ristekdikti).

**Acknowledgments:** We thank the Deutsche Forschungsgemeinschaft (DFG, German Research Foundation) and the Ministry of Research, Technology and Higher Education (Ristekdikti) for providing the funds to conduct this study. We thank Ristekdikti for providing the research permit for field work (No. 322/SIP/FRP/E5/Dit.KI/IX/2016). We thank our field assistants Fahrozi, Erwin Pranata, Syahbarudin and Kairul Anwar for great support during the field campaigns. Thanks to all 'EFForTS' colleagues and friends in Indonesia, Germany, and around the world.

**Conflicts of Interest:** The authors declare no conflict of interest.

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
