# Peer review of "Predicting Tree Sap Flux and Stomatal Conductance from Drone-Recorded Surface Temperatures in a Mixed Agroforestry System—A Machine Learning Approach"

_remotesensing, doi:10.3390/rs12244070_

Round 1
Reviewer 1 Report
This seems to be clear and careful work, and well reported. You should state whether stomatal conductance was measured for only one leaf surface or summed for both (and, if just one, was this done by assumption, or tested?). Please mention the successional status of the tree species considered - do they contrast with oil palm? That might explain their low stomatal conductance vs. oil palm.
Author Response
Dear Reviewer,
Thank you for taking the time to revise our manuscript. We welcome your comments and think they have helped to further improve our manuscript. Please find our point-by point response (in blue color) in the attached word file.
Sincerely,
Florian Ellsäßer

Reviewer 2 Report
This study presents new results and methods to compute ecophysiological variables from remote sensing data. The main objective of this study was to compare different methods to estimate stomatal conductance and sap flux at the tree scale (classical statistical approaches, machine learning, neural network). Results are quite promising at least for predicting sap flux and show that an approach based on random forest algorithms is the most accurate. Even if it remains unclear what is the genericity of these conclusions (low number of trees, limited duration of the experiment) I found that the study opens new perspectives for using such a kind of method in agronomical or ecological context.
I found some of the parts of the manuscript difficult to follow for non-experts in machine learning approaches and that some methodological aspects need to be further described. Finally, the discussion section is too long and does not provide enough information to evaluate the genericity of the results.
Below some detailed comments:
L56-57: Many studies used thermal indexes or data coming from multispectral or hyperspectral cameras to compute proxies of plant transpiration or functioning. What are the main limitations of such approaches? It is thus not clear what are your expectation with your new methods considering these previous works.
L60-61: Could briefly describe here what are the main differences between the different ML algorithms?
L118: What are the variations of the meteorological variables during this two-week period? Adding this information could be very useful to further investigate the genericity of your approach. For instance, some predictions could be of lower quality in some specific meteorological conditions. But you never investigated this possibility in your study.
Table 1: DBH is not defined.
L134: What is the range of variation in stomatal conductance or sap flow during a given day (also an important information in relation with the genericity of your model).
L146: Could you add in supplementary material, some additional information about the time of data acquisition and the associated meteorological variables.
L150: TIR is not defined.
L144: How many spectral bands did you record between 7.5 and 13.5 µm? Could you briefly describe the methods used to « convert » the thermal spectrum into temperatures?
L166: Could you be more precise here? Which kinds of analyses did you perform, what are the key metrics?
L176: Surface temperature (or LE) are considered in other studies as proxies of leaf hydraulic behaviour. It is quite strange to use such kinds of very detailed variables as input variables? Another approach could be to use raw variables (eg spectral bands) extracted directly from images, only, especially for your neural network approach?
L173-182, Table 2. You explain you used 96 variables but we do not find this number in table 2.
L199 : here you have 42 variables but the number of variables in table 2 is still lower.
L196 : how did you split your training set ? Randomly? Will the results be different if another splitting had been done? To which extent? Could you evaluate that?
L240-242, L211-212, L251-253: should move in the introduction section
2.4.4. As far as I understood neural network (or deep learning) are very efficient because they can “compute” their own features from raw variables in order to predict output variables? In your case, I have the feeling you already used complex/computed features as inputs. What could be the consequences of such an “a priori” selection of features? But maybe I am wrong and I did not get everything on ML and deep learning methods…
2.6. You did not perform any model validation on independent datasets (other fields, type of trees?). This could be one limitation to evaluate the genericity of your model. Otherwise, the genericity of your model could be limited to your field and to your two-week period? Could you at least discuss that aspect?
L195-Figure 2. Not clear to me if you separated your dataset depending on species in your calibration procedure? I do not find this information in the M&M section.
L267. Accuracy/MAE/MAPE… It could be interesting to explain further how we can interpret the values of all these indicators? In other terms, what is the rationale of using these four (with R²) different indicators? Adding the formula for these three indicators could help.
The title of section 3.1 (“Prediction accuracy”) is quite ambiguous. Indeed, accuracy refers to one indicator used to assess model performance, but in this section you also show other “metrics”.
Figure 2. How can you get accuracy values equal to 0? Is that a matter of model convergence?
Figure 3. I am not an expert of these methods, but how can you have for the same dataset an accuracy equal to 0% and MAE which are not so bad. Is accuracy computed on the calibration set and MAE on the validation set? I am a little bit lost here with the meaning of the different indicators? You could probably improve the understanding of your results if you clearly answer to my remark for L267.
L340: It is the effect of the variables which is “homogenous”
3.3. Model comparison should move after the section “Prediction accuracy”. Not sure if you need two different titles because in part 3.1. you also compare model performances. I also find Figures 5 (R² values)/9/10 (confidence interval on intercept and slope) and 11 (correlation between observed and estimated variables on a specific case) quite redundant. Could it be possible to reduce the number of figures or use some of them as supplementary information?
4.1. Could you give more precise information on the other methods that could be tested? (e.g sentence L427 is quite obvious…).
4.2 this part is very long, could you reduce it?
4.3 once again, I do not understand the rationale of having 4.3 after 4.2
L543. The statement here is contradictory with L.389-392.
Overall, I found the discussion sometimes too long and quite difficult to follow.
Author Response
Dear Reviewer,
Thank you for taking the time to revise our manuscript. We welcome your comments and think they have helped to improve our manuscript considerably. Please find our point-by point response (in blue color) in the attached MS Word document.
Sincerely,
Florian Ellsäßer

Reviewer 3 Report
The manuscript describes the implementation of machine learning techniques as a standard method to extract and sense physiological activity- in this case stomatal conductance/transpiration and sap flux - from trees.
The idea of extracting information by advanced statistical algorithms has caught new attention in recent years with advance in computational power as well as the introduction and practice in machine learning (supervised and unsupervised) techniques.
This manuscript is well crafted in this respect and deserve to be heard and published.
I have only two minor comments for the authors -
1. previous work published this year (Vitrack-Tamam, 2020), show already a relation between an ensemble method +ANN and stomatal conductance measurement with an AP4 system (Almost identical to this work, with one of two differences is working with a different organism). The other difference is the method of remote acquisition. Please discuss the differences between the two findings, maybe the fact that you worked with trees that are much more complex organisms were the reason for not receiving a good correlation.
2. CWSI - Crop Water Stress Index (Jackson, 1981) is a well established remote sensing index for measuring stomatal conductance in plants by thermal imagery. It uses two references (wet/dry) in order to correlate transpiration and thermal measurements. Please add a short review of this method in the introduction and discuss (in discussion) how this method can improve your findings, model and research in the future.
Citations are good, English can be improved.
A very nice work, Good luck !
Author Response
Dear Reviewer,
Thank you for taking the time to revise our manuscript. We welcome your comments and think they have helped to further improve our manuscript. Please find our point-by point response (in blue color) in the attached MS Word document.
Sincerely,
Florian Ellsäßer

Round 2
Reviewer 2 Report
I would like to acknowledge the authors for the huge piece of work they performed to improve the manuscript. Many parts of the manuscript are now clearer. They also tried to reduce its length to improve its impact. Although the genericity of the approach still needs to be proved I think that this paper represents an interesting first step that could be published in its current form.